# Resurgence, Populism, and Politics 'From Below' †

Keith Cherry

Centre for Global Studies, University of Victoria, P.O. Box 1700 STN CSC, Victoria, BC V8W 2Y2, Canada; kcherry@uvic.ca

† This paper is based on a talk given to the University of Victoria's Center for Global Studies on 10 February 2021. I am grateful for the input participants provided, and all errors remain my own. I would also like to acknowledge the generous finical support of the Killam Foundation, the Center for Global Studies at the University of Victoria, and the Center for Constitutional Studies at the University of Alberta.

**Abstract:** Populist politics are an increasingly prominent feature of contemporary politics around the world. In settler colonies, Indigenous resurgence is also an increasingly important feature of political contestation. Both discourses involve questions of peoplehood, pluralism, and collective agency. The goal of this paper is to explore these phenomena side by side, and ask what they reveal about the present political conjuncture. I argue that both political projects involve a constructive element, as actors build spaces of political contestation beyond the state. In this way, each movement involves an often overlooked contest between politics 'from above' and 'from below'. Ultimately, I conclude that the above/below distinction reveals important cleavages that are obscured by the traditional left/right distinction that structures much political analysis.

**Keywords:** populism; resurgence; politics from below; Wet'suwet'en

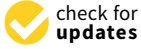



## 1. Introduction

Though the precise definition of the term remains contested, there is wide scholarly agreement that the current surge in populist politics around the world is a defining feature of the present political conjuncture. Case Mudde (2004) has called it the 'populist zeitgeist', whereas Chantal Mouffe (2018) terms it the 'populist moment'. Increasingly, politics is coming to revolve around a perceived conflict between dominant elites and institutions on the one hand, and 'the people' on the other. This struggle raises complex questions concerning who controls dominant institutions and why, who 'the people' are, how they relate to pluralism[1], and how they can express their agency more fully.

Situated here on the West Coast of Turtle Island[2], however, one can scarcely begin to understand the present political conjuncture, the growing problematization of dominant elites and institutions, or even the contested conceptualization of 'the people' without engaging with Indigenous resurgence as a political project. As Indigenous and Settler peoples alike grapple with the ongoing colonization, dispossession, and disempowerment of Indigenous communities, they too are facing profound questions of institutional bias, contested identity, pluralism, and collective agency.

Questions of peoplehood, pluralism, and political agency are central to both populist and resurgent discourses. The goal of this paper is to begin to think of these two phenomena together—to map the family resemblances between them, and to explore the questions they jointly raise. Ultimately, I contend that both discourses gesture towards an important fissure between politics 'from above' and 'from below', each of which provides its own account of peoplehood, pluralism, and political agency. I contend that this distinction between politics 'from above' and 'from below' can make a crucial contribution to our understanding of the current political conjuncture and the struggles that define it.

In Section 1, I take up the 'Shut Down Canada' movement in solidarity with Wet'suwet'en land defenders, using the movement as a window into colonization and Indigenous resurgence more generally. In Section 2, I argue that Indigenous resurgence can be read as,

among other things, a rejection of state-managed pluralism, and an affirmation of a form pluralism based on independent Indigenous authority that rests alongside, rather than beneath, the claims of the state. In Section 3, I explore populism as a contested term, taking up both monist and pluralist conceptions of populist praxis. In Section 4, I draw on Jakeet Singh to argue that both populism and resurgence, each in their own ways, involve a struggle between politics 'from above' and 'from below'. Finally, I take up the prominent tendency to categorize political phenomena as either 'right' or left', and argue that it unhelpfully conceals the distinction between 'above' and 'below', contributing to a post-political moment which both misunderstands contemporary politics and limits the political imagination moving forward. I conclude that the above/below distinction makes a crucial contribution to left/right analyses, allowing for a more nuanced account of contemporary political struggles, their relationship to pluralism, and the points of connection, tension, and friction between them.

## 2. The Historical, Legal, and Political Context of the 'Shut Down Canada' Movement

### 2.1. The Macro-Historical Context

For decades, Wet'suwet'en hereditary leaders have been asserting jurisdiction over their lands, and refusing consent for pipeline developments. In January 2019, and again in January 2020, Canadian police raided Wet'suwet'en territory and forcibly opened their lands to pipeline construction. In response, hereditary leaders called on their supporters to 'Shut Down Canada'. Supporters responded with a nation-wide series of rail, port, and road blockades and occupations, unlike anything in recent memory. The Wet'suwet'en's conflict with the Canadian state, and the 'Shut Down Canada' movement it gave rise to, are interventions in a longstanding history of conflict between Indigenous peoples and the colonial state. Understanding these interventions therefore requires contextualizing current events relative to the basic contours of colonial history more broadly. The following section provides a very brief account of that history, beginning with early-contact interactions, and moving through to the present day.

The practices evident in early-contact interactions between Indigenous and Settler peoples suggest that neither party expected sovereignty over the other[3]. Both European empires and Indigenous confederacies were typically pluralist political systems, with multiple actors wielding independent authority in a system characterized by overlapping claims and negotiated decision-making (Darwin 2011, p. 3). In this context, Indigenous and Settler peoples alike were largely able to understand one another as independent authorities, and engaged in negotiations to establish how those authorities would interact[4]. Thus, Settlers and Indigenous peoples developed a wide variety of treaty arrangements designating the jurisdictional reach of each actor, and establishing hybrid judicial structures and joint decision-making forums which served to regulate the relationship over time[5].

As time passed, however, four important shifts occurred. The first two shifts were ideological. First, European political thought and practice began transitioning away from imperial structures with multiple independent sources of authority, and consolidated into centralized nation-states which understood themselves as monopolizing all authority within their territories[6]. Second, European thought began to take up the concept of race and racial hierarchies (Tully 2018). Together, these ideologies created a discourse that elevated centralized European states over supposedly less-developed pluralistic societies.

The third and fourth shifts were material. With the fall of New France, Great Britain became a dominant military power on the continent (Miller 1989, especially chp. 5). At the same time, the Settler economy was transitioning from a fur- and fish-based economy, premised on cooperation with Indigenous peoples, to an agricultural and staples economy, premised on the displacement of Indigenous peoples from their lands[7].

As a result of these broadly concurrent shifts, Settlers began to re-understand their relationship with Indigenous peoples in ways which both fit with their changing conceptions of race and sovereign authority, and also facilitated the large-scale dispossession of Indigenous lands and resources. Where Indigenous peoples were once seen as indepen-

dent authorities whose relationship to the Settler state could only be structured through negotiation, Settlers now began viewing Indigenous peoples as possessing a lesser claim to authority that could be superseded by the sovereignty of supposedly more advanced colonial states. As a result, Settlers came to see Indigenous governments not as independent authorities, but as sub-components of the British Empire, such that any autonomous authority they held existed at the pleasure of, and subject to unilateral regulation by, the Crown[8].

Indigenous peoples resisted these shifts fiercely, continually centering their status as independent authorities, and their concomitant right to negotiate the terms of their relationship with the Crown (Cherry 2020, esp. chps. 2–4). Settlers, however, pressed their material advantages, and the *de facto* relationship gradually transitioned away from a form of pluralism where independent authorities negotiate their interrelations, moving imperfectly towards a system of 'state-managed pluralism' in which the state attempted to determine the extent and nature of Indigenous authority and its relationship to the Crown unilaterally. Unsurprisingly, Settler authorities used this self-proclaimed prerogative to dispossess Indigenous peoples of their lands, deny and criminalize their traditional governance structures, and violate long-standing treaties and alliances[9].

To take a few illustrative examples, through the *Indian Act*, Settler governments created elected band councils and began interfacing with them, rather than with existing Indigenous governance structures[10]. In so doing, they re-positioned Indigenous authority as a creature of Settler statute, rather than an inherent right. This allowed Settler legislatures to attempt to determine the extent and content of Indigenous authority unilaterally. Similarly, Settler courts began re-interpreting treaties not as nation-to-nation agreements, but rather as a commitment made by a sovereign to their subjects—essentially a policy statement or campaign promise with no legally binding force (Promislow 2014). Once again, this allowed Settler governments to try to dictate the treaty relationship unilaterally. Likewise, inter-communal criminal disputes, which had long been the subject of negotiation, began to be mediated by imperial tribunals and later by ordinary Settler courts (Walters 1995), further positioning Indigenous peoples as subjects of Settler authority, rather than possessors of their own authority.

Indigenous peoples have never accepted these shifts, and many continue to operate their own governance structures[11], insisting on the binding nature of their treaties[12], and contesting the jurisdiction of settler courts to this day[13]. At the same time, Indigenous peoples have made pragmatic use of the band council system, Settler courts, and a host of other, state-managed processes to creatively increase their political room for maneuver, and pursue the interests of their communities. This participation in state structures has not, however, generally entailed abandoning claims to independent authority based on inherent rights that pre-date the state. Rather, Indigenous peoples have expressed their agency within, against, and alongside state structures, often all at once.

The meta-narrative of the historical relationship between Indigenous and Settler authorities is therefore one of profound contestation, as Settlers seek to transform a multilateral relationship between two independent sites of governing authority into a unilateral relationship between a Settler sovereign and its subjects, while Indigenous peoples employ diverse and multi-modal tactics to resist this incorporation, and insist on nation-to-nation negotiations.

### 2.2. Reconciliation as State-Managed Pluralism

The historical record is therefore clear—Indigenous nations were, and were recognized as, independent authorities, and then, the Crown began exercising sovereignty over them without any clear legal foundation beyond racism and military might. Recognizing these facts, Canadian courts have ruled that the pre-existing authority of Indigenous peoples and the *de facto* authority of the Crown must be 'reconciled'[14]. To that end, Canadian courts have elaborated a doctrine of Aboriginal Rights—in essence, if an Indigenous nation can demonstrate that it enjoyed certain legal privileges before contact, the court will seek to

express those privileges as modern rights, and will strike down any government legislation that violates those rights without due justification[15].

The creation of Aboriginal rights has done important work blunting, and calling attention to, many of the worst excesses of colonial rule, and many nations have used them creatively to contest the colonial relationship. However, Aboriginal rights fall squarely within the logic of state-managed pluralism. Indeed, their entire structure presumes that Settler courts will determine whether a right exists, what its content is, whether or not it has been violated, and what remedy, if any, is appropriate—all unilaterally, without any negotiation with, or input from, Indigenous authorities, except in their capacity as plaintiffs in a Settler court.

Courts have also made clear that they will not countenance any Aboriginal right that is incompatible with Crown sovereignty[16], and will not recognize broad rights to self-government or jurisdiction[17]. Even where Aboriginal rights are found to exist, they can be violated by the Settler government if the government can 'justify' the infringement to the satisfaction of a Settler court[18]. Though courts have found that Settler governments must consult with Indigenous authorities in some situations[19], such consultations almost never require actually obtaining the consent of the nation in question. Once again, Settler courts determine how much consultation is necessary, and decide when and how the Crown can override Indigenous objections.

In all these ways, Aboriginal rights are a sort of auto-limitation—a constraint that Settler authorities place on themselves, within the logic of their own unilateral institutions. They do not fundamentally challenge the notion of Settler sovereignty over Indigenous peoples—they simply structure the exercise of that power. The Court's attempt to 'reconcile' Indigenous and Settler authority therefore serves to re-create the basic framework of Settler unilateralism. In the context of Aboriginal law then, the word 'reconciliation' takes on a specific meaning associated with continued unilateralism and the ongoing denial of independent Indigenous authority.

### 2.3. Modern Treaties as State-Managed Pluralism

Even though they facilitate a process of unilateral Settler rule, court judgements on Aboriginal rights also frequently encourage the parties to reach negotiated solutions. To that end, Settler governments have initiated a modern treaty process. Like Aboriginal rights, the modern treaty process has blunted many of the worst effects of colonialism, and opened up new spaces for Indigenous agency. However, it too re-creates the basic logic of state-managed pluralism.

Indeed, the state sets the terms of treaty negotiations unilaterally, deciding what issues are on the table and what sorts of results are possible (Christie 2013, at note 11). Unsurprisingly, Settler governments have chosen to define the scope of the treaties so that they do not meaningfully challenge Crown sovereignty, private property, or capital accumulation. For example, where modern treaties recognize Indigenous jurisdiction, they often include paramountcy clauses, under which Settler legislation prevails in the case of conflict. In other cases, modern treaties include equivalency provisions, where Indigenous law is only valid where it is essentially equivalent to existing Settler laws. Even where Indigenous jurisdiction is not so fettered, Indigenous governments must still operate within the confines of the Canadian *Charter*, and their decisions are reviewable in Settler courts.

In all these ways, modern treaties continue to position Indigenous peoples as a sub-component of the Canadian state, and therefore subject to Canadian sovereignty, rather than recognizing them as independent authorities with a right to negotiate all aspects of their relationship multilaterally[20]. Indeed, the fact that Settlers continue to exercise unilateral authority and deny Indigenous jurisdiction where no treaties exist shows that the state is only willing to countenance Indigenous authority *within* the framework of the Canadian state.

Andrew Woolford's thorough analysis of the BC Treaty Process provides a helpful diagnosis (Woolford 2004; See also Woolford 2001, 2006). Woolford draws on Nancy Fraser's

distinction between two types of transitional justice—transformative repair, which seeks to transform the underlying structures of inequality, and affirmative repair, which seeks to redress historical wrongs without actually transforming the ongoing underlying structures that produce them. Insofar as the BCTC, and the modern treaty process in general, seek to maintain Settler sovereignty and re-present Indigenous authority as a subordinate sub-component of the Settler state, they represent an exercise in affirmative repair.

### 2.4. UNDRIP as State-Managed Pluralism

In part because of the perceived shortcomings of domestic jurisprudence and policy, the *United Nations Declaration on the Rights of Indigenous Peoples* (UNDRIP) is becoming an increasingly prominent part of discussions on Indigenous-Settler relations. In many ways, UNDRIP brings important attention to, and provides potential resources against, the ongoing effects of colonization. However, UNDRIP is the product of an international system which recognizes the sovereignty of Settler states, and which denies Indigenous peoples membership in the international community (Barsh 1983).

Indigenous nations are not represented in the UN's General Assembly or Security Council, and they do not cast votes within the UN system[21]. In order for UNDRIP to be legally binding, it must be voluntarily ratified and implemented by Settler states. Thus, it is Settler states who were in the position to determine the content of UNDRIP, and they are in a position to sign it or not, and to implement it or not, according to their own pleasure. It is not surprising then, that UNDRIP unequivocally affirms the sovereignty and territorial integrity of Settler states. What UNDRIP does is codify a series of voluntary commitments by the sovereign about how they will treat their Indigenous subjects.

Notably, UNDRIP does not require the consent of Indigenous peoples to become binding law. It is emphatically not a multilateral document. Rather, like Aboriginal rights and the modern treaty process, it represents a voluntary auto-limitation by the Settler state. In this way, it serves to retrench and legitimize a unilateral relationship between Settler and Indigenous authorities.

## 3. 'Reconciliation Is Dead': Indigenous Resurgence and the Rejection of State-Managed Pluralism

### 3.1. Resurgence and the 'Turn Away'

Coulthard (2014) argues that the limited rights, financial incentives, and forms of jurisdiction held out by the jurisprudence of Aboriginal rights and the modern treaty process can actually work to help legitimize and perpetuate Settler unilateralism[22]. After centuries of systemically impoverishing and oppressing Indigenous nations, Settler governments offer limited forms of support and autonomy that those nations can scarcely refuse. When Indigenous nations participate in such mechanisms, this is held up as proof of consent, thereby legitimizing Settler unilateralism.

In response to these colonial techniques, many Indigenous academics, activists, community organizers, and traditional leaders have begun advocating a 'turn away' from the Settler state[23]. Rather than focusing their energies on seeking recognition of their authority from the state through court cases or modern treaties, advocates of the 'turn away' focus their energies on enacting their sovereignty directly, without seeking state recognition. This alternative project is broadly known as 'resurgence'.

By emphasizing traditional systems of law, governance, production, and thought, such strategies work to lessen material and psychological dependence on the Settler state by building autonomous Indigenous capacity on the ground (Corntassel 2012). By refusing engagement with the state, such strategies also deprive the state of the ability to legitimize itself through Indigenous participation.

Thus, the 'turn away' works to continually dramatize the dubious historical and legal foundations of Crown unilateralism by enacting the persistence of Indigenous authority. In this sense, the resurgence constitutes a rejection of the politics of reconciliation and the

state-managed pluralism for which it stands, affirming instead the forms of pluralism that characterized earlier phases of the encounter.

### 3.2. 'Reconciliation Is Dead'

The experience of the Wet'suwet'en, of course, has its own specificities, but can, in many ways, be seen as a microcosm of this larger encounter. As in most contact zones[24], the relationship began when the Settler military position was insecure and the Settler economy depended on close Indigenous cooperation to access furs and resources. In this context, the colony's first Governor signed a series of treaties with the Indigenous peoples who lived closest to intensive settlement, thereby recognizing Indigenous nations as independent authorities engaged in a negotiated relationship (Vallance 2015). As time went on, the Settler position became more secure, and the economy transitioned towards gold, lumber, agriculture, and industrial fisheries which depended to varying degrees on the dispossession of Indigenous lands, waters, and resources[25]. At the same time, confederation talks raised the idea of a Settler Canadian nation-state in the place of the imperial system in 'British North America'. As a result of these shifts, and the prevailing racism of the time, Settler authorities in British Columbia stopped seeking treaty relationships, and began establishing Indian reserves and band councils unilaterally (Harris 2003, especially Part 2). The Wet'suwet'en therefore never signed a treaty ceding, or in any way modifying, their jurisdiction and governing authority, yet were nevertheless subject to unilateral assertions of authority from the Settler state. Like Indigenous peoples across the continent, many Wet'suwet'en have contested these assertions fiercely, and continue to do so today through a variety of tactics.

One approach has been to explore the avenues of state-based redress. In the 1990s, Wet'suwet'en chiefs, along with their allies the Gitxsan, brought a groundbreaking legal challenge asserting 'ownership and jurisdiction' over their lands[26]. In 1997, the Court conceded that neither nation had ever ceded its legal authority, but found that the only remedy available came in the form of Aboriginal rights and title, rather than ownership and jurisdiction[27]. In 1994, the Wet'suwet'en sought a negotiated solution instead through the modern treaty process[28]. Here too, however, the state insisted that Wet'suwet'en governments must be subject to Crown sovereignty, and offered to recognize only limited forms of authority. Like many nations, the Wet'suwet'en treaty negotiations stalled, and in 2008, the nation withdrew from the process[29].

Alongside these attempts to engage with the state, the Wet'suwet'en have also expressed their agency outside of the state. In addition to state-funded band councils which administer various reservations on Wet'suwet'en territory, the Wet'suwet'en continue to operate their own, independent governance system, and enforce their traditional laws and occupy their traditional territories, demonstrating their capacity to 'turn away'. In 2009, Wet'suwet'en members erected the Unist'ot'en camp—a healing lodge and cultural center situated directly in the path of several proposed pipelines. Camp members established a border check-point to regulate access to their territories[30]. By denying access to pipeline workers, they enacted their ability to refuse consent without seeking state recognition. In 2018, the Gidimt'en checkpoint was established to support these efforts. By creating spaces of collective action and self-determination outside of state structures, these camps help build governing capacity independent from the state. Such spaces entail not only a negative project of fighting to remove colonial repression, but also a (re)constructive project of (re)building capacity within traditional governance structures which have been deeply disrupted by, and interpolated by, the colonial encounter[31]. In both these ways, spaces such as Unist'ot'en work to refuse the proposition that the state can exercise unilateral authority over Wet'suwet'en people and lands.

Wet'suwet'en members therefore exercise agency both within and beyond state structures[32]. Indeed, the controversy over proposed pipelines has illuminated the complex, and sometimes conflictual, relationship between these tactics. When government and industry sought consultation with The Wet'suwet'en, they were met with mixed responses.

In exchange for certain benefits and concessions, five of the six Wet'suwet'en band councils signed agreements with the project (Bellrichard and Barrera 2020), exercising a form of agency within the state-managed consultation structure. Prominent hereditary chiefs, however, have opposed all pipeline proposals on the basis of traditional Wet'suwet'en law, and put their support behind the Unist'ot'en and Gidimt'en checkpoints[33]. This multi-modal response set off nation-wide debates over who can properly claim to speak for the Wet'suwet'en people (e.g., Wilson-Raybould 2020).

Pipeline proponents, touting the support of band councils, sought injunctions against the camps, and in January 2019, and again in February 2020, heavily militarized RCMP officers raided Wet'suwet'en territory, arrested Wet'suwet'en members, and forcibly opened their lands to pipeline construction without their consent (Wilson 2020b). In response, hereditary leaders called on supporters to 'Shut Down Canada', and supporters responded with an unprecedented, nation-wide series of rail blockades, highway blockades, port blockades, and (re)occupations of public space, including a weeks-long (re)occupation of the BC legislature building (Wilson 2020a; McIntosh 2020; Indigenous youth occupy B.C. 2020; Wikler 2020). Many of these actions were led by Indigenous nations, most notably the Mohawk[34], as an assertion of their own independent authority and indeed, of their own independent diplomatic relationships with the Wet'suwet'en. Other actions involved large numbers of Settlers. The slogan that emerged from these actions was sweeping and direct: 'reconciliation is dead'. Placed in historical context, this claim can be read as a rejection of state-managed pluralism and Crown unilateralism, and a re-assertion of independent Indigenous authority and negotiated pluralism.

Indeed, the emphasis of the anti-pipeline forces is not merely on the inadequacies and injustices of the state, but rather on enacting and revitalizing Wet'suwet'en law, governance, and culture as an independent site of authority. In this sense, the 'Shut Down Canada' actions can be understood as a form of what Robert Odawi Porter (2006) has called 'tribal disobedience'—acts which, from the point of view of the state, constitute 'civil disobedience' or law-breaking actions. Yet, from the point of view of hereditary leaders and their supporters, these same actions appear as necessary to uphold, and are legitimized by Wet'suwet'en law and authority. Rather than law-breaking actions, blockades and other direct actions become law-upholding or law-enforcing actions under an alternative site of legal authority. Hence, action participants consistently refuse to characterize themselves as 'protestors', using terms like 'land defender' instead, so as to characterize their actions relative to Wet'suwet'en, rather than Canadian, law. In short, tribal disobedience dramatizes the co-existence of multiple authorities which are independent in practice, rejecting the politics of reconciliation and enacting an extant alternative.

The 'Shut Down Canada' movement, and the larger current of resurgent politics in which it partakes, can therefore be understood, at least in part, as involving two contrasting political projects. Whereas reconciliation focuses on leveraging accommodations, rights, payments, and governance powers within the settler state in a way that is compatible with the sovereignty of that state, resurgence focuses on strengthening independent legal, political, cultural, and economic institutions which posit a sovereignty all of their own[35].

As I hope is clear, this analysis does not imply a simple dichotomy where a legitimate, resurgent hereditary system confronts an illegitimate, conciliatory band-council. As the Wet'suwet'en demonstrate, these systems and their respective strategies can co-exist in the context of complex and internally pluralistic struggles, sometimes complementing one another, and sometimes occasioning divisive conflict within the community. Indeed, many of the same actors engage in both council-based tactics of reconciliation and hereditary-chief-led tactics of resurgence in complex ways that defy simple dichotomization or binary accounts of legitimacy. Rather, this analysis suggests that the presence of (at least) two intertwined tactical repertoires, and the ability to map the points of connection, tension, and friction between them is crucial to understanding the present political conjuncture.

## 4. The 'Populist Zeitgeist'

The same questions of collective identity, collective action, and orientation towards the state animate populist discourses as well. Firm definitions of populism are an elusive quantity, and academic debates rage about which movements can be considered properly 'populist'. In his thoughtful contribution to this volume, Webber calls populism a 'quintessentially contested concept'. The loose scholarly consensus, to the extent that it exists, positions a movement as broadly 'populist' if it partakes in a discursive binary which divides the population into two groups, a virtuous 'people' and a corrupt 'elite' which rules unjustly over them[36]. However, accounts of what further characteristics follow from these initial premises diverge significantly. I present two such accounts here: a broadly 'monist' understanding of populism, and a broadly 'pluralist' understanding.

### 4.1. Monist Understandings of Populism

What I will call 'monist' understandings of populism generally assume two further characteristics flow from the elite/people dichotomy. First, populism is presented as employing a homogenizing view of the people. The people are presented as internally united and possessing a single, coherent will. In order to maintain this self-image, populist movements actively seek to create the homogeneity they claim, often by excluding racial, religious, or political others from the scope of 'the people'. Thus, monist accounts take populism to be inherently anti-pluralist—pluralism threatens the authority claims populists make, and therefore, becomes something to be both denied and actively eliminated.

Second, as a result of these strategies of homogenization, populist movements are able to portray themselves as representing the will of 'the people' in its entirety, and to characterize dissent as parochial interests opposed to the interests of the people (Levy 2020). This allows populists to present political opposition or institutional constraints as inherently illegitimate. As a result of these two features, populism is often understood as a threat to democracy insofar as it is premised on an active denial of diversity and pluralism, and an active disregard for institutional norms, political opposition, and constitutional fetters.

### 4.2. 'Pluralist' Understandings of Populism

Though the monist account is perhaps the dominant popular understanding, a wide range of populist scholars and practitioners have long pointed to the existence of a range of heterodox populisms which challenge traditional accounts of populism as anti-pluralist.

For example, Boyte (2020) discusses an alternative populist tradition growing from anti-fascist movements, civil rights movements, and the community organizing tradition. His account emphasizes their populist character, but also stresses an intersectional account of peoplehood. Drawing from the Popular Front against Fascism, Boyte argues that the civil rights movement and the Industrial Areas Federation (IAF) developed a common front against oppressive institutions, but understood themselves as inherently plural. Moreover, to the extent that such groups drew binary concepts (fascist/anti-fascist, racist/anti-racist, etc.), they were also deeply informed by a non-violent ethos and a set of non-violent tactics which were oriented towards the eventual conversion of fascists into anti-fascists (and racists into anti-racists, etc.). Thus, the self-understanding of the movement was internally intersectional[37], and the line dividing the movement from its others was understood as fluid such that the goal was not exclusion or expulsion, but conversion and inclusion. Boyte also stresses the fact that this tradition emphasizes the construction of non-state, alternative institutions that are directly democratic and participatory. Though none of the movements eschew engagement with the state, they understand state institutions as one site of struggle among many. Indeed, Boyte problematizes the conception of politics which places all political activity within the state, and presents the alternative tradition he describes as working to open up non-state forms of direct rule and participatory problem solving (Boyte 2020, especially p. 65).

Similarly, Illuzzi (2020) argues forcefully that the Illinois Black Panther Party (ILBPP) and its 'rainbow coalition' must be understood as populist according to a variety of criteria, yet he presents the movement as a multiethnic, politically diverse coalition built on a conception of common, but differentiated, oppression. According to Illuzzi, this allowed the coalition to build what he terms a 'moderate story of peoplehood' where the people are seen as composed of many groups, each of which is entitled to a significant degree of self-determination, yet none of whom expect unilateral sovereignty. Instead, this moderate story of the people envisions an internally diverse people continually negotiating amongst itself. Illuzzi also emphasizes the permeability of boundaries between 'the people' and the oppressors, noting that moderate groups were sometimes excluded from 'the people' even as new groups were continually brought in (Illuzzi 2020, especially at 79–81). In fact, the groups involved devoted significant energy to educating white communities, building critical awareness among gang-members, and spreading breakfast programs as a means to convert erstwhile enemies into supporters. Boundaries between self and other were therefore dynamic: the goal was not the elimination of the other, but rather their education, conversion, and inclusion. Indeed, Illuzzi emphasizes the centrality of the ILBPP's free breakfast programs and its ability to provide other social services directly to the people as a critical part of the movement's strategy. Though the BPP did run candidates in elections, its focus was on constructing alternative institutions that were participatory in nature. Ultimately, the coalition's goal was to educate the people 'to the point that they can run things themselves'[38].

Ouziel (forthcoming a, forthcoming b) emphasizes similar features in his account of Spain's 15 M movement and the associated party, Podemos. Though many key theorists of European populism, including Chantal Mouffe and Boa de Sousa Santos, see 15 M and Podemos as the epitome of a state-centric left populism, Ouziel contends that 15 M can only be properly understood through careful attention to its constructive project. Though individuals being 15 M do sometimes support candidates and even political parties such as Podemos, they are also engaged in building non-state, directly democratic, assembly-based institutions designed to facilitate direct governance by the people themselves. Though the construction of 'the people' does take place against a construction of 'the elite', the people are understood as internally diverse, with significant attention paid to differences in regional identities, immigration status, gender, and political ideology[39]. Moreover, a focus on non-violent and educational tactics aims at the conversion, rather than elimination, of Spanish elites.

Similarly, numerous commentators on the 'Occupy' movement stress the construction of alternative institutions, and of the participatory, open assembly in particular, as the defining feature of the movement (Mathews 2019; Kinna et al. 2019). Critics have argued that the movement's framing of the 99% vs. the 1% is inherently homogenizing, and belies a politics of representation whereby a privileged few claim to speak for a great many, particularly people of color, women, and people in the global south (Mathews 2019). In several encampments, these tensions became explicit, as in the Decolonize Oakland and (Un)Occupy Albuquerque movements (Fortier 2017; Miller and Davis 2012). Though attempts at more inclusive occupations were not always successful, the consistent contestation of inclusion does show that 'the 99%' was not understood in an undifferentiated manner and indeed, that participants recognized that giving expression to intersectional interests was a necessary component of the political project. Mathews (2019) contends that the representative claim inherent in the 99% framing was in explicit tension with the anti-representative, participatory institution building of the movement. He contends that this tension was managed in at least two ways. First, the imperative to 'occupy' envisioned an ever-growing number of public assemblies that would eventually encompass the planet, and as such, was inherently inclusive. Second, the use of open public assemblies allowed anyone to wander in and participate as an equal. Together, Mathews argues these techniques worked to blur the border between inside and out, and also, to express an inclusionary logic that seeks to bring diversity into the movement, rather than to suppress or deny it.

Together, these movements display a set of characteristics which, to varying degrees, challenge monist constructions of populism and, in particular, its relationship to pluralism and state institutions. In contrast to dominant accounts, these movements represent a tradition of populism in which: (1) the people are understood intersectionally rather than homogenously, and discursive and political strategies are geared towards inclusion, rather than exclusion or elimination; (2) the movement embraces a selective anti-institutionalism that challenges the institutions and elites of the state, even as it works to support alternative institutions and leadership positions.

Importantly, this tradition is not merely *compatible* with pluralism, it is *inherently pluralistic*. To the degree that it involves constructing multiple, decentralized non-state institutions, it also necessarily involves constructing relationships between those institutions. While the state exists, it also means defining relationships with the state. Indeed, this family of populism involves the construction of distinctly populist forms of pluralism. Ouziel's work, for example, focuses on the processes of 'joining hands' through which individuals being 15 M connect with one another, with allied organizations and parties, and with the state itself, in ways that remain rooted in their own political and ethical modalities (Ouziel forthcoming b, especially pp. X, 8–16). In so doing, they enact a form of pluralism that follows its own logics distinct from those of state pluralism. Indeed, the process of joining hands constitutes a multifaceted strategy to de-center the state and parochialize it by placing it alongside, and in relation with, a variety of participatory non-state structures—in other words, theirs is a distinctly populist form of pluralism.

The alternative tradition of populism revealed by Boyte, Illuzzi, Ouziel, and others therefore illuminates both a pluralist form of populism, and also a populist form of pluralism. Bretherton (2012) calls this tradition 'political' populism and contrasts it with 'anti-political' populism, where anti-political populism requires the denial of pluralism through homogeneous understandings of the people for the purpose of state-capture, and properly political populism works to create a capacity for pluralist negotiation within the people in their capacity as civic citizens.

## 5. Populism, Resurgence, and Politics 'From Below'

### 5.1. Populism and Resurgence

As we have seen, efforts to define populism as a phenomenon are troubled by the contest between monist and pluralist accounts. Monist accounts emphasize strategies to create a homogeneous conception of the people, and to represent that conception through a single figure for the purpose of state-capture. This account is inherently anti-pluralist and, because it allows actors to claim to speak for the people in their entirety, can easily be mobilized to undermine any institutions, norms, or practices which fetter state power. It is a top-down discourse which concerns itself primarily with control over, and maximizing the use of, state power.

However, there are also pluralist traditions of populism in which the people are understood intersectionally, rather than homogenously; discursive and political strategies are geared towards inclusion, rather than exclusion or elimination; and the movement embraces a selective anti-institutionalism that challenges the apparatuses of the state even as it works to support alternative institutions at the grassroots level. This tradition is a bottom-up project concerned primarily with the capacity of the people to govern themselves in ways that can include, but also exceed, the state.

Similarly, the 'Shut Down Canada' movement involves, among other things, two intertwined accounts of pluralism. 'Reconciliation' signifies a state-led process defined by the use of rights and delegated powers to constrain the sovereignty of the Settler state and open spaces of Indigenous agency without calling the state's monopoly on political authority into serious question. This account revolves around a monist view of political power as state power, and attempts to circumscribe pluralism within state boundaries in order to maintain that monist account. It is prone to authoritarian tendencies because the state, in claiming to speak for the people as a whole, also claims the authority to set the

terms of its relationship with Indigenous peoples unilaterally in a classic example of the tyranny of the majority.

In contrast, 'resurgence' emphasizes a plurality of independent sources of political authority, potentially including, but not limited to, the state. Many actors in Indigenous resurgence movements oppose a corrupt Settler elite, and view the institutional and constitutional structures that uphold its privilege as illegitimate and non-binding[40]. As in pluralist accounts of populism, however, the understanding of 'the people' operative in this conflict is decidedly not homogenous. This is true in at least three ways. First, because of the colonial context of the struggle, 'the people' are disaggregated into Settler and Indigenous groups, and indeed, into particular Indigenous nations. An understanding that Indigenous and Settler peoples are differently positioned relative to structures of social privilege and oppression is fundamental to the movement's practice and discourse. Second, the understanding that Settlers and Indigenous peoples are differently positioned extends more broadly, recognizing that gender, sexual identity, sexual orientation, class, race, and other factors are also relevant, implicating different structures of privilege and oppression, and necessitating different responses in order to bring about good relationships. Third, invocations of 'the people' are often both intergenerational and inter-species, in that the object of political representation includes both ancestors and future generations, and also entails a recognition that non-human forms of life are 'plain citizens' in the ecological order (Tully forthcoming; Leopold 1969, pp. 237–64, 239–40). The opposition between the people and the elite does not therefore result in a homogenizing picture of the people, but rather produces an inherently intersectional, disaggregated conception of the people.

Both cases then, feature a commitment to grapple with what Tuck and Yang (2012) have called "understandings of uncommonality". As Tuck and Yang note, such understandings work to continually "un-coalesces coalitions", building friction around the representation of identity into the very practices by which the community is created and maintained.

Moreover, though the 'shut down Canada' movement, and others like it, are clearly premised on the rejection of the authority of Canadian elites and a challenging of the legitimacy of the Canadian state, they also revolve fundamentally around a valorization and revitalization of traditional leadership and law. As Odawi Porter makes clear, such movements are not simply anti-institutional, they are also alter-institutional, in that they posit alternative sources of authority, and accept alternative constraints on political action[41].

In so doing, both projects work against what Swain (forthcoming) has called 'concretization" —an ongoing process to imbue the status quo with a sense of permanence or inevitability. For Swain, concretization proceeds by encouraging subjects to express their agency through established channels, and this is achieved at least in part though rendering alternatives forms of agency invisible. By enacting alternative forms of social organization, Indigenous resurgence works to deconcretize Settler colonialism. Likewise, we might say that the ILBBP, Occupy, 15M and others have worked to deconcretize liberal democracies in their own ways. Thus, in embracing an alter-institutional element, both populism and resurgence act "like a brick through the Overton window" (Henderson forthcoming), expanding the scope of political contention beyond the confines of statist politics.

### 5.2. Understanding the Current Conjuncture as a Contest of Projects from Above and Below

Both populist politics and decolonial politics therefore involve both an inherently monist, statist, and top-down political project with anti-pluralist and potentially authoritarian tendencies, and also a non-state, horizontal, and radically pluralist political project[42]. I contend that this distinction maps helpfully onto what Jakeet Singh (2015) calls 'recognition from above and below'.

Politics 'from above' begins from the premise that the state monopolizes legitimate governing authority, and focuses on the degree to which the state does, or should, recognize the collective agency of other groups as a part of its larger attempt to secure the liberty of its citizens. In so doing, approaches 'from above' ensure that pluralism takes a form which

is purely internal to, subordinate to, and defined unilaterally by, the state. Singh contrasts this approach with 'recognition from below'. Approaches 'from below' begin from the premise that multiple collective agents can make legitimate claims to authority, without recognizing any special priority for the claims of the state. Rather than treating the state as *the* authority and centering the degree to which it accepts or rejects competing claims, such approaches position the state as merely one authority among many, and center negotiation and contestation between the state and other authorities. As a result, political strategies 'from below' are not necessarily state-centric, but can also work to change the relative balance of power between authorities by strengthening the capacity of non-state actors.

As Singh argues, accounts 'from above' tend to adopt a states-eye view of politics, and concern themselves primarily with how state institutions do, and should, exercise power. As such, they tend to view citizens in an undifferentiated light, and prize universal logics and standards. Similarly, monist accounts of populism, which we might call 'populism from above', adopt a state's point of view. They focus on control over, and the potential undermining of, state institutions, and revolve around the universalizing (and thus, homogenizing) binaries necessary for electoral success and state-capture. State-led 'reconciliation' similarly focuses on universalizing rights discourses as a mean to legitimize the monopolistic authority of state institutions.

Conversely, the pluralist tradition of populism, which we might now call 'populism from below'[43], is not unconcerned with the state, but its institutional politics are inseparable from the creation and valorization of non-state institutions. These movements take an intersectionally disaggregated view of the people, and envision an inherently pluralist democracy that de-centers state structures. Likewise, resurgent movements proceed from an inherently disaggregated understanding of the people, and revolve fundamentally around the resurgence of traditional, non-state institutions, worldviews, and lifeways.

Of course, as Singh rightly points out, many complex movements draw on strategies 'from above' and 'from below' at the same time (or intermittently) (Singh 2015, p. 68). The Wet'suwet'en and their supporters are clear exemplars of such an approach, as are the movements studied by Illuzzi, Ouziel, and Boyte. Hunt and Holmes (2015) call for a 'both/and' approach to such binaries. Likewise, Ouziel emphasizes the need to study the diverse variety of ways that political actors blend these approaches together in practice (Ouziel forthcoming a, forthcoming b). In this way, the above/below distinction is not a rigid formal binary. Rather, the above/below distinction allows us to map a repertoire of practices, and their orientations towards the state and towards pluralism, as important dimensions of contemporary political praxis. In particular, the distinction allows us to map the points of contact, tension, and friction between the two projects, which, I contend, produces a richer and more nuanced account of the present political conjuncture.

### 5.3. Centering the Above/Below Distinction

I contend that this distinction between politics 'from above' and 'from below' makes at least three important contributions to our understanding of the present political conjuncture.

First, the above/below distinction reveals that the popular distinction between right and left belies a deeper commonality. Both left and right populism, as Mouffe conceives them, share a set of statist commitments. In essence, right populism works to define the elite/mass binary in such a way as to produce electoral advantage and facilitate right wing policy goals, whereas left populism does the same in service of left policies[44]. Though Mouffe lauds some non-state populist movements, she ultimately assimilates their value to electoral contestation and state-capture[45]. Both then, are essentially electoral strategies. Mouffe does stress that left populism must rely on a 'chain of equivalences' between different struggles, and thus, allows for a disaggregated conception of 'the people'. To this degree, left populism embraces a liberal form of pluralism that may be absent in right populism. However, even left populism, as Mouffe conceives it, places pluralism firmly *within* the state, implicitly eschewing the idea of pluralist relationships beyond the state[46]. Moreover, left populism, like right populism, remains committed to a politics of

representation which allows, and indeed requires, that 'the people' be represented by a single figure or party operating through a single set of institutions (the state), therefore introducing the homogenizing and anti-procedural tendencies that critics of populism worry about.

Comparing 'left' and 'right' political parties' stances on Indigenous sovereignty reveals a similar common ground. Insofar as all mainstream parties embrace a society where the state enjoys a monopoly on political authority, capital retains free access to Indigenous lands, and Indigenous empowerment is envisioned as receiving greater influence in state and corporate structures, they all share a common set of commitments to politics 'from above'[47].

In both cases, the contest between left and right takes place entirely within a politics 'from above'[48]. Using the left/right distinction as a lens can therefore limit our analysis, rendering approaches 'from below' invisible, or assimilating their value to statist politics. In so doing, the left/right binary concretizes a political imaginary dominated by competing projects of homogenization and representative politics, erasing the constructive dimensions of many contemporary political movements, the more radical pluralism such constructive projects involve, and the reimagined forms of democracy they entail. Such an understanding of contemporary politics dramatically misunderstands the present conjecture, and misses the profound challenge it poses to traditional political structures.

This is all the more striking because Mouffe presents left and right populism as competing responses to an era of 'post-politics' where center-left and center-right parties have become nearly indistinguishable (Mouffe 2018, pp. 10–15). Yet from a grassroots point of view, the commitment to state-capture makes both right and left populists essentially indistinguishable. Seen in this light, the contest between right and left creates a 'post-political' moment of its own, where our conception of politics is confined to politics 'from above'. Richard Day (2005) refers to this as 'the hegemony of hegemony', noting that both left and right frames encourage a view that political aspirations can only be realized through state power.

By grouping left and right populism together as both proceeding 'from above', the above/below distinction allows us to shift our conceptual focus from what differentiates right and left movements to what ties them together, and thus, re-casts the political, revealing a deeper statist consensus that cross-cuts Mouffe's proclaimed antagonisms and opening space to engage with projects 'from below'. This not only renders visible a whole range of under-studied political phenomena, it also reveals that practices of concretization and deconcretization constitute a major terrain of political struggle today. The current conjecture is, in many ways, not just a contest between left and right, but a contest between state and non-state actors over control of the Overton window itself.

By decentering the state as a theatre of contestation, and drawing our attention to alternative governance structures, the above/below distinction also centers a different tactical repertoire. Alongside electoral contestation, forms of prefigurative politics, civil or tribal disobedience, non-violent, and even armed, contestation have become central to the political repertoire not merely as ways to contest elite rule, but as ways to enact extant alternatives. In an impressive study of contemporary activism, Day lists no fewer than six forms of non-hegemonic action: "dropping out of existing institutions; subversion of existing institutions, through parody; impeding existing institutions, via property destruction, 'direct action case work', blockades, and so on; prefiguring alternatives to existing institutions, often via modes of activity that otherwise fall within the purview of a hegemonic politics, for example protests; and finally, construction of alternatives to existing forms that render redundant, and thereby take power from, the neoliberal project". The above/below distinction makes these non-hegemonic practices legible on their own terms, rather than assimilating their meaning and value to strategies of state-capture. In particular, this helps bring into view the constructive dimensions of political movements.

Second, by expanding the field of view to include not only projects 'from above', but also those 'from below', the above/below distinction centers the question of how

these two projects relate. Such questions arise both within movements and between them. Indeed, the cases studied here show that many movements blend approaches from above and below in practice. This raises questions about how actors are weaving their projects together, what frictions these attempts generate, and what possibilities they create or foreclose for specific movements, struggles, and groups. Similarly, important questions arise around how distinct projects 'from below' and their respective efforts to deconcretize contemporary politics work to either foreclose or make room for one another. For instance, populist invocations of popular sovereignty can often be strikingly ahistorical, implicitly erasing Indigenous sovereignty. For this reason, populism has often come into conflict with decolonial movements, as the (un)Occupy struggles demonstrate. In essence, the alternative future opened up by populist discourse risks foreclosing the alternatives opened by resurgence. In other cases, resurgent activists and settler anarchists are struggling to co-articulate mutually supporting futurities in which the de-concretization performed by each only opens new room to maneuver for the other (Day 2001; Fortier 2017; see also Swain forthcoming). In opening room for such inquiries, the above/below distinction allows for deeper understandings of contemporary politics, and also a richer understanding of pluralism itself. Ouziel's work, for example, explores no fewer than six ways that projects 'from above' and 'from below' interrelate in contemporary Spain[49]. Such work offers a richer understanding of the Spanish conjuncture and, crucially, of the possible futures it may generate.

Third, as Singh explains, being attentive to accounts "from below, in particular, is crucial because it sheds light on the praxis of those most marginalized by, and often most resistant to, dominant norms and institutions—those whose hardships, practices, capabilities, insights, and movements for change are easily and routinely ignored, overlooked, misunderstood, or underestimated. Top-down forms of politics often proceed from monological and monistic theories that have not been informed by the realities and perspectives of those at the bottom, but are nevertheless imposed on them. The worst off, then, end up bearing the greatest burden and having the least say over the reshaping of their lives and communities. This is dangerous not only because it ignores the needs, hardships, insights, and knowledges of those most affected but also because it polarizes by excluding and rendering incomprehensible the political claims and actions of many subalterns and counter-hegemonic social movements. This narrowing and obfuscation of the political field serves both to reinforce the hegemony of prevailing norms and to conceal an important source of insights" (Singh 2015, p. 69). Indeed, by limiting our attention to projects 'from above', the left/right distinction not only further marginalizes the voices of oppressed communities, it also privileges the voices—and interests—of the corporate and political elites which dominate statist politics. In this way, the right/left distinction is a deeply conservative lens at precisely the moment when political projects 'from above' and their partisans in the dominant classes are posing an existential challenge to life on earth as we know it, through unchecked climate change and ecological collapse.

In all three of these ways, shifting from viewing politics through a left/right lens to an above/below lens can shed light on aspects of the present conjuncture which are otherwise marginalized, invisibilized, or dismissed, thereby unveiling more transformative political possibilities. This is not to suggest that the above/below distinction can replace the left/right distinction entirely as a way of making sense of contemporary politics. Left and right remain useful markers in analyzing the current conjuncture. Indeed, just as projects from above can be both left and right, so too can projects from below[50]. The two axes are therefore crosscutting—by analyzing contemporary phenomena using both lenses, we can understand contemporary movements more precisely, and map their connections and differences more richly.

## 6. Conclusions

In an effort to better understand the current political conjuncture, this paper has explored the concurrent phenomena of populist politics and Indigenous resurgence. Using

the 'Shut Down Canada' movement as an example, I have framed Indigenous resurgence as a contest between 'reconciliation', understood as a state-based, inherently monist project, and 'resurgence', understood as an enactment of deep, non-state pluralism. Turning to populism, I have explored the debate between monist and pluralist understandings of populist politics. I have argued that, despite their differences, both of these contests can be usefully captured by Singh's distinction between politics 'from above' and 'from below'. Finally, I have argued that the above/below distinction is of crucial importance in understanding the present political conjuncture, the nature of current political movements, and their relationship to pluralism.

Specifically, I contend that this distinction is a helpful corrective to a left/right distinction because it allows us to see the constructive dimension of many contemporary struggles. In so doing, it allows us to imagine political possibilities beyond the state, centers the question of how these projects relate to statist politics, and encourages us to center the rhetorical, tactical, and aspirational repertoires of marginalized communities. In all these ways, the addition of an above/below distinction illuminates a wider current of political action than the left/right distinction can alone, facilitating a richer understanding of contemporary politics, and a fuller appreciation for its radically transformative potential.

**Funding:** Not applicable.

**Data Availability Statement:** Not applicable.

**Conflicts of Interest:** The author declares no conflict of interest.

## Notes

1      I am grateful to the reviewer who suggested exploring the work of William E. Connolly as a resource for the account of pluralism developed here. Connolly's corpus is extensive and varied, and a thorough engagement will have to be the subject of future essays. For now, let me acknowledge the connection, and note that readers may find Connolly's pluralism of interest.

2      The term 'Turtle Island' is drawn from the occurrence of the turtle in many Indigenous creation stories, including the Anishinaabe and Haudenosaunee. It is commonly used to refer to North America while implicitly calling into question the European prerogative to name, govern and exploit a land which was already occupied and governed when they arrived. (Snyder 1995, pp. 454–62).

3      "Aboriginal people did not infer or accept a relationship of domination, nor did the Europeans, in practice, try to impose one". Canada; (Erasmus and Dussault 1996, p. 120).

4      For an extended discussion see Cherry, *Practices of Pluralism* especially chapter 2.

5      For a dsicussion of the development of shared and hybrid structures at the political and legal levels respectively see Cherry, *Practices of Pluralism*, chp. 2 and 3.

6      Tully helpfully discusses this transition from the "ancient constitution" models, with their polyvocal theories of authority to the "victorious modern language of constitutional uniformity" in which political authority speaks with/through one voice. (Tully 1995, pp. 58–59).

7      For example, a memorial address given to Prime Minister Laurier by Shuswap, Okanagan and Nlaka'pamux chiefs stressed the difference between fur-trade Settlers and agricultural Settlers, noting that the former were economically interdependent and therefore cooperative and neighbourly, while the later wanted only to displace Indigenous peoples. (Tully 2017). More generally see (Miller 1989, chp. 2). For a detailed account of this transition see (Walters 1995).

8      See e.g., Cherry, *Practices of Pluralism* at pp. 51–53.

9      See Cherry, *Practices of Pluralism* chps. 4, 3 and 2 respectively.

10      (Miller 1989, p. 105–222). See also (Truth and Reconciliation Commission of Canada 2015; Abele 2007).

11      See for example Unist'ot'en Camp, "Governance Structure" https://unistoten.camp/about/governance-structure/ (accessed on 1 May 2021).

12      See for example, the recent surge in treaty-activism in L'nu territory. Melkita'n, "Colonizers Being Colonizers: Lobster Fishing & the Continued Oppression of L'nu'k in Mi'kma'ki" *Yellowhead Institute Policy Briefs*, No. 92 (2021); (Cherry 2020).

13      To take a few illustrative examples: in *R v. Francis*, the defendant claimed that Canadian law did not apply to members of the Mohawk Nation on reserve because they are not Crown subjects and because no treaty established such jurisdiction. The court's reply was succinct "even if there existed some basis on which to challenge the acquisition of sovereign jurisdiction by Canada, the domestic or "municipal" courts of Canada lack competence to question it". *R. v. Francis*, ONSC 05/17, [2007] 85 OR (3d) 45 at 7. In *1536412 Ontario Ltd. v. Haudenosaunee Confederacy Chiefs Council*, ONSC 08/82, [2008] CanLII 28041, the court held that "the special rights enjoyed by the aboriginal communities under our constitution do not include sovereignty" at 22. In *R. v. Ignace*, [1998] 103 BCAC 273, [1998] 156 DLR (4th) 713 at 11, the appellants argued that Canadian jurisdiction did not

apply on lands which had never been acquired by treaty. Citing *Delgamuukw*, the court found that any First Nations sovereignty, jurisdiction or legality failed to survive the application of common law. In *R. v. Day Chief*, 2007 ABCA 22, [2008] 412 AR 29, the defendants argued "that Alberta lacks jurisdiction to regulate their conduct—and indeed any conduct on the specific land where the offences are alleged to have occurred—because that land was acquired by fraud". Citing *Sparrow*, the court held that underlying sovereignty was always with the Crown, even on lands where no treaty had been signed.

14    For various expressions of the 'reconciling' function of aboriginal rights, in chronological order, see e.g., *R. v. Sparrow*, [1990] 1 SCR 1075, [1990] 70 DLR (4th) 385 at 39; *R. v. Van der Peet*, [1996] 2 SCR 507, [1996] 4 RNLR 177 at 31; *Delgamuukw v. British Columbia*, [1997] 3 SCR 1010, [1997] 153 DLR (4th) 193 at 81; *Haida Nation v. British Columbia*, 2004 SCC 73, [2004] 3 SCR 511 *at* 20. For a discussion of the legitimizing roles of reconciliation see (Simpson and Le Billon 2021).

15    *R. v. Bernard*, 2005 SCC 43, [2005] 2 SCR 220 at 51.

16    "aboriginal interests and customary laws . . . were absorbed into the common law as rights, unless (1) they were incompatible with the Crown's assertion of sovereignty, (2) they were surrendered voluntarily via the treaty process, or (3) the government extinguished them" *Mitchell v. M.N.R.*, 2001 SCC 33, [2001] 1 SCR 911 at 10.

17    *R. v. Pamajewon*, [1996] 2 SCR 821, 138 DLR (4th) 204 at 27.

18    See e.g., *Sparrow* at 1114-1115; *R v. Gladstone*, [1996] 2 SCR 723, [1996] 137 DLR (4th) 648 at 62–65, and *Delgamuukw* at 160–169.

19    *Haida*, at 27.

20    For further discussion see (Mack 2010; Alfred 2000). I would also like to acknowledge that Taiaiake Alfred has recently resigned from the University of Victoria for presiding over a toxic and patriarchal environment, and has faced numerous criticisms regarding his relationship with women in particular. It would be wrong to cite to his work without acknowledging his problematic behaviour and making clear that I in no way condone his actions. I stand in solidarity with all the students and others who have experienced harm at Alfred's hands and I strive never to recreate such harms in my own life or work. To ignore Alfred's undeniable contributions to the field or refuse to cite to his work would, however, erase not only Alfred, but also the people he has harmed and the work they have done to seek accountability. Instead, I prefer to foreground the conflict. In so doing, I hope to draw attention to the enormous work that our society, and our educational institutions in particular, must do to confront patriarchy and other forms of oppression on an ongoing basis. See e.g., Anna Dodd, "Founding Director of UVic's Indigenous Governance program says he "embodied toxic masculinity" The Martlet, 7 March 2019, accessed 10 October 2021 at https://www.martlet.ca/professor-taiaiake-alfred-resigns-from-uvic/.

21    For a complimentary analysis of UNDRIP see (Asch 2019).

22    (Coulthard 2014, especially chps. 1 and 2). Coulthard does not discuss UNDRIP but complementary accounts can be found in Barsh, "International Law" and Asch, "UNDRIP".

23    For a discussion of 'the turn away' in the thought of Taiaiake Alfred and Leanne Simpson, See (Coulthard 2014, pp. 154–55).

24    Instead of thinking of a contact as a singular event occurring in a singular place (when Settlers met Indigenous peoples for the first time on the East coast), Mary Louise Pratt offers the more useful idea of a "contact zone"—a shifting space of contact that moves over time as Settler populations expand and encounter different nations. See (Pratt 1992, especially p. 4).

25    For a discussion of changing economies in BC see (Lutz 2009).

26    *Delgamuukw v. British Columbia*, [1997] 3 S.C.R. 1010, [1997] 153 DLR (4th).

27    *Delgamuukw* at 7. For discussion see (Borrows and Rotman 2012, p. 245). Note also that the issue of whether title had been proven was sent to retrial, leaving the Wet'suwet'en without even this.

28    BC Treaty Commission, "Wet'suwet'en Hereditary Chiefs Table" accessed 5 March 2020 https://www.bctreaty.ca/wetsuweten-hereditary-chiefs#:~:text=Wet\T1\textquoterightsuwet\T1\textquoterighten%20traditional%20territory,%2C%20Tsimshian%2C%20and%20Lake%20Babine.

29    Unist'ot'en Camp, "Timeline of the Campaign" accessed 5 March 2020 http://unistoten.camp/timeline/timeline-of-the-campaign/.

30    Unist'ot'en Camp, "Background of the Campaign" https://unistoten.camp/no-pipelines/background-of-the-campaign/ (accessed on 1 May 2021).

31    For discussion of the need to positively re-build Indigenous law, see the work of Val Napoleon generally. e.g., (Napoleon 2012).

32    Compare Ouziel's discussion of "one foot in one foot out' strategies and Tully's five types of civic activity. (Ouziel forthcoming b, p. 79; Tully 2009, especially p. 280).

33    (Proctor 2020). In fact, three Matriarchs who supported the project were stripped of their hereditary titles over the matter, (Baker 2020).

34    Of the Haudenosaunee confederacy.

35    For a considerably more nuanced account of resurgence, reconciliation and the manifold relationships between them see (Tully et al. 2018).

36    Ibid. See also (Albertazzi and MacDonnell 2008; Müller 2016; Laclau 2007).

[37] Or at least intersectionalizing- as we will see, many of the movement under consideration continue to struggle to become more intersectional and pluralist over time. Intersectionality is not, in this sense, a static attribute, but rather an ongoing and always imperfect process.

[38] ILBPP leader Bobby Seale as quoted in (Illuzzi 2020, p. 75).

[39] For a series of examples of how 15M grappled with a variety of intersectional concerns, esepceally regarding gender, regional identity, and migration status see (Ouziel forthcoming b, especially p. 71, 104, 119–20, 165, 167).

[40] As a result, politicians, media and academics have portrayed the Wet'suwet'en and their supporters as threats to the rule of law and indeed, to democracy itself, portraying the movement in much the same way that populist movements are often portrayed. For thoughtful discussion of this point see e.g., (Seatter and Turner 2020; Gunn and McIvor 2020; Hyslop 2020).

[41] For example, traditional Wet'suwet'en leaders laid down a set of 'supporter protocols' that allies are asked to conform to. See Unist'ot'en Camp, "Supporter Protocols" https://unistoten.camp/supportertoolkit2020/ (accessed on 1 May 2021).

[42] In fact, this affinity may help explain, in part, why the call to "Shut Down Canada" received such an unprecedented level of support from Settler allies across the country. Because the resurgence of Wet'suwet'en law and governance proceeds 'from below', it may have natural resonance with Settlers who are also engaged in populist contestation from below, creating fertile ground for widespread solidarity in this 'populist moment'. Indeed, Swain stresses the complementarity between Settler politics 'from below' and Indigenous resurgence movements. (Swain forthcoming).

[43] Several authors have taken up the term 'populism from below', though few have used it in a sense comparable to the one developed here. For instance, Andrade uses the term in reference to economic class, where populism 'from below' embraces policies which are in the interest of the poor and varieties 'from above' are in the interest of the elite. Similarly, Richards uses the term to signal parties who leaders are from the lower classes, rather than the upper classes. Brentin and Trošt discus populism 'from below' as a sort of academic orientation, where studies 'from above' focus on formal political institutions and studies 'from below' explore the pop cultural elements of populism. Of the authors surveyed, only Tarragoni uses the term in way that is comparable to the sense developed in this paper. In their exploration of Chavist populism, Tarragoni distinguishes between the top-down populism of the state, and it's counterpart in the grassroots self-organization of poor communities. Here, populism 'from below' bears the same participatory, disaggregating/inclusionary, and constructive dimensions stressed in my own account. See (Andrade 2020; Richards 1981; Brentin and Trošt 2016; Tarragoni 2019).

[44] (Mouffe 2018, p. 18). For a similar account of Mouffe to my own, and one to which I am greatly indebted, see (Ouziel forthcoming b, especially Chapter 4 and the Conclusions).

[45] "I think the Indignados and Occupy had commonalities in their rejection of institutions, political parties and trade unions. Theirs was a purely horizontalist perspective and I think it missed the fact that building hegemony must necessarily pass through the state. I am not defending in any way that politics is limited to the parliamentary sphere. The horizontal dimension is very important, but to have a real impact and transform things a vertical element needs to be articulated. Its objective being one of *'becoming* state'." (Mouffe forthcoming, p. 105).

[46] "Some theorists say that the problem is with representative democracy *per se*. Following from this, they suggest that the solution to the current crisis is the elaboration of models of direct democracy. I see it differently, I think that the problem of our crisis of representation is that our societies are not representative enough; there are numerous sectors of the society that do not have a voice. This is, I think, a consequence of our democracies no longer being agonistic. When people think there is no left and right anymore, then there are no alternatives. Therefore, what we need to do in this conjuncture is to re-establish partisanship". Chantal Mouffe, "Agonistic Representative Democracy" at 109. She continues on 112: "To put into practice a pluralist democracy one needs representative institutions to give an institutional form to pluralism. This is why I think political parties are key if we want to have an agonistic democracy".

[47] For a discussion of the mismatch between Indigenous demands and statist responses see (Day 2001).

[48] This is not to deny that there are meaningful differences between a right wing government and a left wing government, both in terms of policy and procedure and in terms of their approaches to diversity and decolonization more specifically. However, both left nor right statists embrace a form of state-managed pluralism, and thus both are potentially prone to authoritarian tendencies. In this sense, both remain committed a basic relation between state and subject which is precisely what politics 'from below' throws into question.

[49] (Ouziel forthcoming b). See a useful table comparing these approaches at X.

[50] For example, the exemplars of politics from below in this paper have been overwhelming left-wing projects. However, the alt-right is increasingly experimenting with political strategies that might be considered, in some sense, to proceed from below. Tracing the contours of such right-wing projects from below, and their relation to pluralism, may challenge some of the conclusions drawn here. This, however, is precisely the point. By viewing such movements as not only right-wing, but also as potentially proceeding from below, we open new conceptual possibilities to understand these movements, their orientation to the state, their orientation to pluralism, and their relation to other movements across the political spectrum.

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
