# Peer review of "Resurgence, Populism, and Politics ‘From Below’"

_socsci, doi:10.3390/socsci10110422_

Round 1

Author Response

I appreciate the reviewer’s kind and supportive words, particularly regarding the style of engagement with Indigenous peoples. I’m also grateful for the helpful suggestion. I briefly outline the changes made in response to each suggestion below.

1) I’ve word-smithed the ‘Centering the Above/Below Distinction’ slightly and hope it now comes off more emphatically.

2) I am very grateful for the suggestion to engage with Richard Day. I had forgotten how much I love his scholarship, and how much I’ve learned from it over the years. I’ve inserted a citation to “Who is this we that gives the gift?” and also engaged with the tactical repertoire he lays out in Gramsci is Dead as well as his concept of the ‘the hegemony of hegemony’.

I am familiar with the means/ends distinction through the word of Gandhi, Tully, Gregg and Mantena but look forward to engaging with Ciccariello-Maher moving forward. However, after some consideration, I decided that you are right to suggest that engaging closely with this distinction in the present article would risk muddying the present argument.

3) I agree entirely that the above/below distinction compliments, rather than replaces a left/right distinction. I’ve made edits throughout to ensure that this is clear, including drawing on Hunt and Holmes’ “both/and” approach.

4) thank you for this correction, I’ve revised the reference to Coulthard to remove any sense that reconciliation is intentionally designed to cultivate psychological attachment.

5) thank you, I’ve noted that it was the judgement that was issued in 97, rather than the case being brought that year.

6) This is the comment I am most grateful for. In previous revisions to this article, I was at pains to ensure that my account didn’t embrace a simple account of band council = bad, hereditary leadership=good, as this is not a view I endorse. Reading the reviewers take on this issue helped me see several ways in which those earlier revisions had been insufficient.  I’ve now a) added text acknowledging that many individuals engage in both systems of governance and both sets of tactics, b) re-framed my portrayal of Unist’ot’en not as a perfect space of hereditary authority, but rather as a space outside of state structures where hereditary systems that have been damaged and interpolated by colonialism can be reimagined and revitalized, c) I’ve acknowledged that resurgence requires more than the removal of colonial repression, but also involves a re-constructive element, and have cited Napoleon on this point.

Altogether, I think the reviewers comments make this article more nuanced and, as a result, make its conclusions more robust. My gratitude to the reviewer for all their help

Reviewer 2 Report

The paper engages with a very important and interesting topic by discussing the conceptual nature of political contest through the distinction between “populism” and “resurgence”. The author explains in a clear and precise way how the politics of populism and resurgence are related to “questions of peoplehood, pluralism and political agency” (line 34) and places her/his analysis in the broader context of the colonial context. By taking as an example the “Shut Down Canada” movement in relation with the Wet'suwet'en land defense, the author develops a distinction between “reconciliation”, which remains caught within the confines of state authority and legal/institutional jurisdiction, and “resurgence”, which helps us articulate a pluralist account of political protest and alternative sources of authority and contestation beyond state sovereignty.

The main argument here is that these two accounts of political protest point to a different direction and that there is an urgent need to rethink the structural conditions of populism through its pluralization. In order to do this, the author mobilizes the distinction between “politics from above” and “politics from below”. While “politics from above” is related to a monist conception of the people and necessarily understands political power through the lens of the state and its institutions, “politics from below” is going beyond the monist account of populism and the understanding of the people as a homogeneous entity through its openness to a plural account of political power and authority. The pluralistic account of populism points to the articulation of an alternative democratic ethos that challenges the legal and institutional prerogatives of the state. The author concludes that the distinction between “politics from above” and “politics from below” expands political imagination and enriches our understanding of contemporary political struggles by going beyond the traditional division between the Right and the Left.  

This is a well-written and readable article. The general structure of the paper is methodologically adequate and the reader follows easily the development of the main argument. The author develops her/his position through a thorough and careful theoretical reconstruction of the subject and provides a vast amount of empirical evidence – taken mostly from the fields of sociology and law – in order to corroborate the development of the main argument from an empirical point of view.

Some problems to be addressed before publication include: a) The distinction between “politics from above” and “politics from below” gives the impression of a rigid and formal distinction. While the author presents this distinction as an alternative to the “left/right” distinction, she/he nonetheless contends that, from a sociological point of view, “many complex movements draw on strategies ‘from above’ and ‘from below’ at the same time (or intermittently)” (p. 17; lines 569-570). In this respect, a number of important questions remain unanswered. How do we make sense of a fascist movement promoting the politics from below? Does not the alt-Right movement in the US – or at least some particular segments of it –promote forms of political power, self-organization, and solidarity going beyond the state? Some clarification is needed here in order to more fully understand the social and political implications of the distinction between politics from above and politics from below; b) the work of the political theorist William Connolly is a very useful resource for this project. Connolly’s concept of “deep pluralism” – the author uses the same expression in order to designate a robust pluralism beyond the state – is very useful in order to understand political authority and social struggles beyond the confines of state sovereignty. The author should address Connolly’s original thoughts on the matter on a few citations.

Author Response

I am grateful to reviewer for their kind appraisal of the article and their succinct synopsis of its contributions.

In response to the reviewer’s first suggestion, I have:

  -emphasized that the above/below distinction is not intended as a rigid, formal distinction

-clarified that the above/below distinction is intended to compliment, rather than replace, the left/right distinction

-acknowledged that just as approaches ‘from above’ can be classified as either right or left, so too approaches ‘from below’, such that it is possible to imagine alt-right projects as proceeding from below.

- acknowledged that the examples I draw on are exclusively from the political left, and that engagement with right/below movements may challenge or extend my account. Ultimately, I have argued that this actually demonstrates the value of the above/below distinction, in that analyzing the alt-right not only as right wing, but also as potentially proceeding from below, makes it possible to analyze the movements’ relationship to the state, to pluralism, and to other contemporary movements more completely.

In response to the second suggestion, I appreciate the encouragement to engage with Connolly, and I read several of his pieces in an attempt to situate my work relative to his.

My initial sense is that Connolly uses the term ‘deep pluralism’ very differently than I do, in that Connolly appears to me to be referencing the ability of fundamentally different cultures and worldviews to exist within a single polity/institutional structure. In my paper, I meant to signify a state of affairs where each group has its own institutions such that pluralism becomes a matter of working across systems, rather than a matter of designing a single inclusive system for all. To avoid any confusion with Connolly’s sense of the term, I removed the term ‘deep pluralism’ from my paper.

My initial sense is that Connolly’s relationship to sovereignty is also very different from my own. Connolly often seems to presents his project as responding to threats posed to state-pluralism either by the association of state-sovereignty with a single nation (see e.g. William E. Connolly “Pluralism and Sovereignty” in Chambers & Carver eds. William E. Connolly: Democracy, Pluralism and Political Theory (London: Routledge, 2008) or by transnational institutions, norms, and capital flows which escape sovereign control (see e.g. William E. Connolly “Beyond the Moral Imperative” in Chambers & Carver eds. William E. Connolly: Democracy, Pluralism and Political Theory (London: Routledge, 2008). In this sense, the project strikes me as an attempt to make sovereignty and pluralism compatible. My own sense is that sovereignty and pluralism are inherently incompatible. My account is therefore more directly aligned with classical pluralists like Harold Laski and G. D. H. Cole, who present state-pluralism as inherently deficient because of anti-pluralist tendencies in the very logic of sovereignty itself.

However, I am aware that Connolly’s body of work is extensive and varied, and I have not had time to cavass it exhaustively. Connolly’s influence is such that I am loathe to offer a casual or careless appraisal. As such, I feel uncomfortable taking a definitive stance on my relationship to Connolly’s work at this time. Ultimately, I decided to respond to the reviewer’s thoughtful suggestion by committing to read more Connolly going forward, and by including the following text in a footnote:

"I am grateful to the reviewer who suggested exploring the work of William E. Connolly as a resource for the account of pluralism developed here. Connolly’s corpus is extensive and varied, and a thorough engagement will have to be the subject of future essays. For now, let me acknowledge the connection, and note that readers may find Connolly’s pluralism of interest. Pending a more in-depth exploration of his work, I will say only that I share many of Connolly’s concerns, including exploring the role of power dynamics in pluralism, and the proliferation of sites of democratic engagement beyond the state."